# Fracture Behaviour of Real Coarse Aggregate Distributed Concrete under Uniaxial Compressive Load Based on Cohesive Zone Model

**DOI:** 10.3390/ma14154314

**Published:** 2021-08-02

**Authors:** Jingwei Ying, Jin Guo

**Affiliations:** 1Department of Building Engineering, School of Civil Engineering and Architecture, Guangxi University, Nanning 530004, China; gj736541632@163.com; 2Key Laboratory of Engineering Disaster Prevention and Structural Safety of Ministry of Education, Guangxi University, Nanning 530004, China; 3Guangxi Key Laboratory of Disaster Prevention and Engineering Safety, Guangxi University, Nanning 530004, China

**Keywords:** concrete, real aggregate, meso-scale, DIC, cohesive, fracture behaviour

## Abstract

Two-dimensional meso-scale finite element models with real aggregates are developed using images obtained by digital image processing to simulate crack propagation processes in concrete under uniaxial compression loading. The finite element model is regarded as a three-phase composite material composed of aggregate, mortar matrix and interface transition zone (ITZ). Cohesive elements with traction–separation laws are used to simulate complex nonlinear fracture. During the experiment, digital image correlation (DIC) was used to obtain the deformation and cracks of the specimens at different loading stages. The concept of strain ratio is proposed to describe the effectiveness of simulation. Results show that the numerical strain ratio curve and stress–strain curves are both in good agreement with experimental data. The consistency between the cracks obtained by simulation and those obtained by DIC shows the good performance of cohesive elements as well as the effectiveness of simulation. In summary, the model is able to provide accurate predictions of the whole fracture process in concrete under uniaxial compression loading.

## 1. Introduction

Concrete is widely used as a building material. The damage research of concrete material has a significant influence on engineering disaster prediction and protection. With the development of computer technology, scholars have found that meso-scale simulation of concrete is very helpful in studying the failure mechanism of concrete, and the macro-mechanic response of concrete is the reflection of its meso-mechanical and micro-mechanic characteristics. Therefore, the meso-scale numerical simulation has become a hot topic in the field of civil engineering in recent years.

Based on the macroscopic experimental phenomena of concrete, several concrete crack models have been proposed to simulate the fracture behaviour of concrete. Among them, the discrete element method (DEM), the discrete crack model and smeared crack model are mainly used at present [1,2,3,4,5]. DEM was proposed by Cundall [6]. It is a kind of numerical calculation method, which is mainly used to calculate how a large number of particles move under given conditions. The accuracy of DEM models depends on how well researchers calibrate the contact model expressions and their parameters [7]. The discrete crack model simulates cracking by defining nodes’ split, which can describe the geometric discontinuity at cracks and inserting zero-thickness elements between solid elements is usually used in this approach to simulate cracks [8]. Classical discontinuous models include lattice model [9,10,11], interface constitutive model [12], etc. More discrete models have been proposed in recent years with the development of numerical simulation, among which the cohesive zone model [13] is widely recognized as a reliable approach for the advantages in evaluating damage [14].

The cohesive zone model is a simple method of modelling the fracture of concrete specimens, with very good agreement with experiments. It is able to adequately predict the behaviour of uncracked structures [15]. Different from traditional fracture mechanics theories, which are inappropriate for application when microcrack and subcritical crack are not ignorable [16]. The cohesive zone model can describe special fracture behaviour caused by microcracks and subcritical cracking because the method uses fracture energy to define element deformation and damage [17].

In the last decade, tremendous efforts have been made in studying evolutions of damage and fracture about concrete. Many scholars have done research about finite element models with random distribution aggregates [18,19,20,21]. However, some research shows that there are limitations in FE models with random distribution aggregates. For example, Xiong et al. [17] indicate that the simulation result changes when the aggregates forms are different.

Recently, simulation by FE models with real aggregates has become a reliable method as digital image processing becomes popular in civil engineering [22,23,24]. Both Sun et al. [25] and Yang et al. [26] established two-dimensional meso-scale finite element models with real aggregates. The former adopted the discrete element modelling method to simulate concrete fracture under uniaxial compression. The latter use phase-field regularized cohesive zone model simulating concrete fracture in uniaxial tension. Both results show the stress–strain curves and the fracture behaviour obtained by simulation are in good agreement with the experimental data. Yue et al. [27] developed two-dimensional meso-scale finite element models with real aggregates to simulate concrete fracture. In the meantime, FE models with random distribution aggregate were also developed. The elastoplastic constitutive model is used in both simulations. The study shows results obtained by the former simulation are more consistent with experimental results. The studies above show the reliability of the meso-scale model with real aggregate. Compared with the real aggregate model, the random aggregate model still has a deficiency.

Some scholars have established FE models with real aggregates, in which cohesive elements were used to model cracks. For example, Trawiński [28] presents two-dimensional meso-scale models to simulate fracture behaviour in notched concrete beams subjected to three-point bending test. Ren [29] developed two-dimensional meso-scale finite element models with real aggregates, cement paste and voids of concrete to simulate concrete fracture under uniaxial tension. Both simulation results show a good agreement with the experimental results. That proved the effectiveness of numerical simulation with cohesive elements.

The cohesive zone model has advantages in simulating cracking. However, very few studies report its application in concrete compression simulation, which has a more complex damage evolution in fracture processes [17]. Concrete compression failure is more common in engineering. Therefore, the cohesive zone model is adopted to simulate compressive fracture of the real aggregate model in this study to provide a new idea for studying mesoscale compression failure of concrete.

In this study, simulations of uniaxial compression tests were carried out. 2D meso-scale FE models with real aggregates are developed for fracture modelling in concrete based on images from digital image processing. Zero-thickness cohesive interface elements with traction–separation constitutive laws were adopted in these models to simulate complicated fracture processes.

## 2. Materials and Test

### 2.1. Experimental Materials

The ordinary Portland 42.5 cement (Fusui Xinning Hailuo Cement Co., Ltd., Nanning, China) was used as cementing material in the specimens. natural river sand (Shiwang Building Materials Co., Ltd., Nanning, China) was used as fine aggregate with a fineness modulus of 3.09. The natural coarse aggregate is granite with a diameter of 5–25 mm. The performance indexes of natural fine aggregate and natural coarse aggregate are shown in Table 1 and Table 2, and the aggregate grading curves are shown in Figure 1. The specific mix ratio of the specimens is shown in Table 3. The maintenance period of all specimens was 28 days under the condition of 20 ± 3 °C temperature and 95% relative humidity.

In this experiment, concrete cube specimens of size 100 mm were cast in the lab. Some of which were used for uniaxial compression tests and the others were used to extract the real aggregate images by digital image processing after polishing.

### 2.2. DIC Program

Digital image correlation (DIC), also known as digital speckle correlation (DSC), is a method of non-contact, non-interference, full-field optical measurement of strain and deformation based on computer image principle, image data processing and numerical calculation [30]. Figure 2 presents the digital image correlation process. Firstly, the camera is fixed with a tripod to avoid image shaking in the environment and focused on the specimen. Then the time of the pressure testing machine is synchronized with the camera’s time so that the subsequent image processing is completely corresponding with the constitutive processing. Finally, the computer program is used to calculate the pictures to obtain strain contours and displacement contours. The camera used here is the industrial camera (model: JHSM1400f, Shenzhen Jinghang Technology Co., Ltd., Shenzhen, China), and it would take about 1000–1500 pictures in a single experiment).

## 3. Simulation Program

### 3.1. Modelling

The actual aggregates images of concrete were obtained by digital image processing [31,32,33], with which 2D FE models were established. The Digital image processing (DIP) procedure is shown in Figure 3.

In order to make the results more reliable, images of aggregates with different geometry and spatial distribution were used in establishing models (RealNC01/NC02). In summary, simulations of two square models with the size of 100 mm were carried out in this study (NC02/NC02). The real image and corresponding model as shown in Figure 4.

Many scholars have established four-phase models to study concrete fracture. The four-phase model means there are four components in the model, which contains aggregates, cement paste, interface elements inside cement paste (CIE) and ITZs [29,34].

However, the model was divided into five parts in this study. Apart from the four components mentioned above, the interface elements inside aggregate (AIE) were appended. All the interface elements are cohesive elements.

There are advantages in appending interface elements inside aggregate (AIEs). Firstly, cracks may appear in the aggregate. Adding AIEs make aggregate fracture in simulation possible; increasing interface elements in the model means increasing the potential crack path of the specimen, which makes the fracture closer to the real situation.

The aggregate elements and cement paste elements are defined as solid elements (CPS3), and cohesive elements (COH2D4) were used to simulate fracture behaviour in this study. Previous research has shown that the models’ deformation and fracture can be well simulated using zero-thickness interface elements [35,36,37]. Therefore, in the numerical simulation, all the interface cohesive elements (COH2D4) are set to zero thickness.

### 3.2. Material Properties

The 2D solid elements (CPS3) for aggregates and cement paste were assumed to behave linearly elastically. For interface elements, linear damage evolution is selected to describe the traction–separation law because of its excellent computing efficiency and high accuracy [38,39] and the quadratic nominal stress criterion is used to assess crack initiation.

Another basic component of the cohesive zone model is the fracture energy, which was used to define damage evolution. There are two modes of fracture energy that need to be defined in 2D FE models, the normal mode (mode I) and the first direction shear mode (mode II). Mix mode (normal mode + shear mode) fractures are more common than pure mode fractures in concrete failure. Therefore, the Benzeggagh-Kenane (B-K) fracture criterion [40] is applied to define the mixed-mode fracture energy. The quadratic nominal stress criterion and mixed-mode fracture criterion are shown in Figure 5.

Due to the lack of experimental data, the material parameters of cohesive elements are set by referring to the researches of Ren [29], Xiong [17], and other scholars [34]. Different kinds of aggregates’ mechanical characteristics were listed in Xiong’s research. So the parameters of interface elements inside aggregates (AIEs) are from Xiong’s study.

For parameters of ITZs and CIEs, there are different settings in different studies. Some previously research consider the mode II fracture energy is about 20 to 25 times larger than mode I fracture energy [17,41]. However, in other studies, the shear fracture properties were assumed to be the same as the normal ones [29,34].

Therefore, based on the studies above, the interface elements’ parameters are calibrated to make the simulation closer to the experiment. The parameters used in the simulation were listed in Table 4.

### 3.3. Mesh Forms and Boundary Condition

Compared with the random aggregates, the aggregates in the real aggregate model are more irregular in shape and more nonuniform in distribution. Moreover, it has other shortcomings, such as the aggregate edge cannot be as smooth as the random aggregate.

Therefore, the mesh cannot be evenly distributed as the random aggregate model. Furthermore, there are some problems with the interfacial mesh between aggregate and mortar, such as the mesh area is smaller than other regions, the density of mesh is higher, and the mesh size is nonuniform. These problems make the real aggregate model difficult to converge because the calculation results are sensitive to mesh distortion. Besides, the interface between aggregate and mortar is the weakest part of the concrete model, where the first crack appeared.

In summary, in the numerical simulation of concrete, the crack propagation and cracking are related to the mesh generation, and different mesh generation forms may lead to different results. Therefore, considering the influence of mesh size on the numerical simulation, four different mesh forms are compared. Since 1 mm mesh size was used in many studies [28,42,43,44], the different mesh sizes were set around 1 mm in this study. The four mesh forms are listed below:The approximate element size is 1 mmThe approximate element size is 2 mmThe approximate size of the cement paste element is 1 mm, and that of the aggregate element is 50 mmThe approximate size of the cement paste element is 2 mm, and that of the aggregate element is 50 mm

In ordinary concrete, the stiffness of aggregate is much greater than that of mortar and ITZs, so the fracture of aggregate is usually not considered. Therefore, in order to reduce the number of mesh and improve computing efficiency, the (3) and (4) forms were set.

The number of solid elements (CPS3) and cohesive elements (COH2D4) are listed in Table 5.

Figure 6 indicates the stress–strain curves of different mesh sizes. It shows that all four curves are converged, and their linear elastic stages almost coincide.

Figure 6 presents that compared with the curve with the element size of 2 mm (blue curve), the curve with the element size of 1 mm (red curve) changes more gently near the peak point, which is in accordance with the damage of the concrete model mainly occurs in ITZs and CIEs. With the increase of cohesive elements, the plastic deformation of the mortar part will be more reasonable and more uniform.

Furthermore, the curve of the element size divided into 50 mm in aggregate and 2 mm in mortar (green curve) is not much different from the curve with the element size of 1 mm (red curve). Compared with the curve of element size divided into 50 mm in aggregate and 1 mm in mortar (black curve), those two curves are smoother, the fluctuation range is smaller, and the convergence is better.

Therefore, considering the balance of computational efficiency and accuracy, the last mesh form was adopted in this study.

In order to improve the similarity between simulation and experiment, the boundary conditions of the model refer to the working principle of the RMT machine. The upper and lower sides of the two-dimensional real aggregate model are respectively connected with an analytic rigid body, the upper rigid body is fixed, and the lower rigid body is loaded upward. The loading speed is set to 40 mm/s, and dynamic explicit (in Abaqus) analysis is used in this model with a step time of 0.01 s.

### 3.4. Parametric Calibration

As mentioned above, there are different settings in different articles about the parameters of interface elements. In this section, different fracture energy ratios (RF) of three internal interfaces (AIE, CIE and ITZ) are discussed. RF is calculated by the following equation:
(1)$$\mathrm{RF}=G_{s}^{C}/G_{n}^{C}$$

The value of normal mode fracture energy is fixed, the value of shear mode fracture energy is changing to varying the ratio. The simulation results of different RF are shown in Figure 7.

RF(A), RF(C), RF(I) represent fracture energy ratios of AIE, CIE, ITZ, respectively. When we change RF(A), which means we only change the value of shear mode fracture energy in AIEs, the other values remain unchanged.

As shown in Figure 7a, the curves of different RF(A) overlap, which declares that the change of RF(A) has no effect on the stress–strain curve as a result of the fracture energy of AIEs is much greater than that of the others.

The stress–strain curves of different RF(C) are shown in Figure 7b. When RF(C) = 25, the peak value of the curve is the largest, which means the peak value of the stress–strain curve increases as the shear fracture energy of CIEs increasing. Also, RF(C) has an effect on the slope of the descending section of the curve. The larger the RF(C) is, the smaller the slope of the descending part of the curve is.

The stress–strain curves of different RF(I) are shown in Figure 7c, which reflects that the peak value of the stress–strain curve increases with the increase of RF(I). RF(I) also affects the slope of the descending section of the curve. Contrary to RF(C), the larger RF(I) is, the larger the slope of the descending section is.

In conclusion, the shear fracture energy of CIEs and ITZs influences the peak value and the slope of the descending section of the stress–strain curve. This conclusion is helpful in calibrating parameters.

## 4. Result Analysis

### 4.1. DIC Result Analysis

DIC contours shown in Figure 8

When the specimen is loaded to 20% of the maximum load (Figure 8➀), several cracks have appeared in the middle of the specimen surface, which forms the rudiment of the first through crack. At this time, the specimen is in the stage of stable crack development.

When the stress reaches 50% of the peak stress (Figure 8➁), the cracks that existed in point 1 are connected to form the first through crack (crack a) on the surface of the specimen. Then the fracture behaviour of crack a is analyzed through displacement contours. The colour of the x-direction displacement contour changes abruptly on both sides of the arrow, which means the crack exists at this place [45,46]. The shape and position of this crack are consistent with crack a. Also, a sudden change in colour takes place at the top of y-direction displacement contour (marked by the circle), the shape of it is obviously the same as the upper part of crack a. In total, the displacement contours indicate that the fracture behaviour of the upper part on crack a is shearing while the lower part is opening.

It can be seen from the strain contour that crack a begins to branch at the place marked by the arrow. At the same position, obvious colour mutation can be seen in both displacement contours, which means the fracture behaviour of this branch crack is shearing. When the crack encounters aggregate during crack development, the crack will stop developing and develop along the other weak surfaces of aggregate and cement matrix [45]. That is the cause of cracks branching.

There is a specific crack in this strain contour, crack b, a transverse crack. The fracture behaviour of crack b is analyzed through x and y displacement contours. It is obvious that there is no crack existent in the x-direction displacement contour. However, in the y-direction displacement contour, the crack exists, and the value in the top right corner of the y-displacement contour is larger than that in other places. In conclusion, crack b is caused by out of plane failure.

When the stress reaches the peak point (Figure 8➃), there are multiple through cracks on the surface of the specimen at this time. In addition to the previous crack a, there is also through cracks c and d. From the displacement contours, it can be seen that cracks c and d are affected by both opening and sliding, which means the fracture behaviour is mainly shearing.

When it comes to the 75% P_max_ after the peak (Figure 8➄), some parts of the strain contour disappeared, as cracks e and f. Comparing the strain contour with the peak one, we can conclude that it is the strain of crack c and d increased to form crack e and f. The disappear parts represent strain exceed peak strain there.

When the stress is reduced to 50% of the peak stress (Figure 8➅), the crack on the surface of the specimen is more significant, and spalling occurs. When the stress is reduced to 20% of the peak stress (Figure 8➆), the specimen is almost crushing, and it is about to lose its bearing capacity. The surface of the specimen has peeled off in an extensive range, and the integrity is destroyed (as shown in Figure 8➆).

All the y-direction displacement contours state that the displacement in the lower part of the contour is larger than that in the upper part, which is consistent with the experimental loading. From all contours, it can be concluded that the failure behaviour of the specimen is mainly shearing, which conforms to the compression failure theory of concrete.

### 4.2. Simulation Result Analysis

The stress–strain curve calculated by numerical model NC01 and the corresponding contours of each stage is shown in Figure 9.

Before peak point, the stress increases with the increase of strain, up to the stress reaches 75% of the peak value, there are some cracks in the strain contour, and these cracks exist at the interface of aggregate and mortar (Figure 9➂). However, the stress contour shows that the stress distribution is uniform and there is no failure element.

When the stress reaches the peak value (Figure 9➃), there are some failure elements in the upper right part of the stress contour, which means initial cracks appear in the model. In the meantime, the strain contour shows that cracks exist, and the number of cracks in the strain contour is larger than the number of failure elements in the stress contour. Moreover, the failure elements are contained in the cracks. In both stress contour and strain contour, the tensile stress near the failure elements is larger than other parts in the model.

After the peak point, the curve turns down, and the stress decreases with the strain increase. The number of failure elements in the stress contour increases with the increase of strain. When the stress comes to about 75% P_max_ after the peak (Figure 9➄,➅), the trend of cracks and the rudiment of main cracks can be seen in the stress contour. The number of cracks in the strain contour is also increasing, and still more than failure elements in the stress contour, but the gap between them is not as big as before. Still, the tensile stress is larger in the regions near the failure elements in both contours.

At the stage of 25% P_max_ post-peak, the slope of the curve is close to zero (Figure 9➆). Compared with the previous point. The number of failure elements in the stress contour increases, but the distribution is almost the same; on the contrary, the number of cracks in the strain contour decreases.

At this stage, the amounts and distribution of fracture elements within the stress contour are both approach to the cracks shown in the strain contour, and the stress distribution tends to be uniform.

After the slope of the curve is equal to zero (Figure 9➇), the number and distribution of failure elements in the stress contour at this point have no obvious change compared with the previous one; the number and distribution of cracks in the strain contour also the same as the previous one. At this time, the number and distribution of cracks in the strain contour are almost the same as those of failure elements in the stress contour and the stress distribution is uniform.

Through the stress contours shown in Figure 9, there are no failure elements at point ➂, which indicates the stress distribution is uniform. However, the stress distribution at point ➃ became nonuniform. That is because some ITZ elements already damage before reaching point ➃, which causes the stress redistribution, also leads to tensile stress concentration around the damaged elements. Then the initial failure elements appear [47]. This is why the stress is larger near the failure elements, while the stress distribution in the area far from the failure elements is more uniform.

Failure elements gradually increase after the peak point. We further observe the phenomenon from the contours of point ➄, ➅ and ➆ that the stress is larger near the failure elements.

However, the model is about to lose bearing capacity at point ➇. So, the stress distribution is uniform and close to zero in the contour.

The strain contours shown in Figure 9 present that the cracks first appear at the interface transition zone. From 75% P_max_ to P_max_, cracks increase rapidly (Figure 9➂,➃). After the peak point, the distribution of cracks in the strain contours almost stay the same, the only difference between the final point (Figure 9➇) and peak point (Figure 9➃) is the number of cracks.

Finally, the stress contours are compared with the strain contours. Before the final point, the number of cracks in the strain contours is larger than the number of failure elements in the stress contours. However, when it reaches the final point, the number and distribution of cracks in the strain contour are almost the same as those of failure elements in the stress contour, which means not all the cracks shown in the strain contours will evolve into the failure elements. As when some of the strain cracks became local failure elements, cracks nearby will become unloaded and close up [29]. That makes the number of cracks decrease.

The development process of cracks can be seen in Figure 10. The damage begins at the ITZ elements between the aggregate and the mortar, and the micro-cracks are connected to form the initial cracks (Figure 10a).

When the stress is small, the cracks occur in the ITZ elements, forming a crack zone surround aggregates (Figure 10b).

When the stress is high, the mortar interface elements (CIEs) begin to damage (Figure 10c), and the cracks are located close to the aggregate crack zone, forming the rudiment of the shear band.

With the loading time increase, the residual strength of the specimen decreases. The number of CIE cracks increase and connected with the ITZs cracks forming the shear bands (Figure 10d). These shear bands divide the specimen into several regions.

As the loading continues, the residual strength of the specimen continues to decrease, and the width of the cracks on the specimen gradually increase (Figure 10e). The stress distribution of the specimen is uniform and close to zero.

After that, the deformation becomes slipping between different regions, and it will cause some mortar regions to spall until the specimen is completely destroyed.

Figure 10f,g are the real failure image of the specimen and the corresponding DIC contour. Comparing them with Figure 10e, it can be seen that the real cracks of the specimen are similar to the result of numerical simulation. This consistency shows that the simulation results are in good agreement with the experimental results. Through this simulation, it can be concluded that the fracture behaviour of concrete under compression is mainly shearing.

### 4.3. Comparison between Experimental and Simulation

In Figure 11, the stress–strain curves obtained from the experiment and simulation are compared. As shown in the figure, the pictures in the solid wireframe are the real concrete section, and the pictures in the dotted line frame are the two-dimensional specimen model.

Experiment 1 curve is obtained from the experiment, where the cross-section image of the sample is shown in the solid black frame. Experiment 2 curve was also obtained from the experiment, and the specimen used in the experiment is represented by the section image in the solid red frame. Simulation 1 curve is the result that corresponded to the specimen model in the black dotted line frame. Simulation 2 curve is the result of the model showed in the red dotted line frame.

The experimental results and simulation results are well corresponded, and the first group of curves and the second group of curves both show a high degree of consistency.

The concept of strain ratio is proposed to further illustrate the agreement between the simulation and the experiment, which is defined as the ratio of the number of crack elements to the total number of elements. If the strain of an element exceeds the reference strain, then the element is considered a crack element.

The experimental strain ratio is obtained from DIC strain contours. As a DIC strain contour is actually a matrix, each pixel on the graph is a corresponding strain value in the matrix, and each value of the matrix is regarded as an element. So, the number of elements exceeding the reference strain were obtained through the matrix, and the corresponding strain ratio of the strain contour was obtained (as shown in Figure 12).

Based on the same theory, the strain values of each element under different frames can be obtained through the simulation result file, by which the number of elements exceeding the reference strain was obtained. Then we got the strain ratio of simulation.

The strain ratio curves of the experiment and simulation under different strain reference values are compared (Figure 13). The solid line is the experimental strain ratio curve corresponding to the left side longitudinal axis. The dashed line is the simulated strain ratio curve, and the corresponding longitudinal axis is the right axis. The transverse axis of the curves is strain.

It can be seen from the figure that the strain ratio corresponding to the experiment and simulation is not exactly the same. The experimental strain ratio is larger than the simulated strain ratio. This is because the experimental strain ratio is obtained by DIC strain contours. DIC strain contours indicate the fracture of the concrete specimen surface. there will be spalling in the surface area. Therefore, the maximum value of the experimental strain ratio curve is close to one, which means the specimen surface is completely damaged. However, in the simulation, the damage of the model is described by cohesive elements, which form certain paths to express cracks. Therefore, the large area damage would not happen in the simulation, so the simulated strain ratio value is small.

The strain ratio curve can reflect the crack development process of the specimen. According to the experimental strain ratio curve, when the strain is less than 0.0002, micro-cracks appear on the surface of the specimen. From the peak strain, the slope of the curve becomes smooth, which means the increasing rate of cracks slows down. The tendency is consistent with the real experimental crack development shown in Section 4.1.

The simulated strain ratio curve shows when the strain exceeds 0.0005, micro-cracks begin to appear on the model. When the strain reaches the peak value, the slope of the curve is still steep. The slope of the curve would not become smooth until the strain was close to 0.0015. The tendency is consistent with the simulation results shown in Section 4.2.

It is concluded from the above that if the experimental and simulated strain ratio curves are in good agreement, that means the failure process of simulation is close to the real one.

Figure 13 shows that when the strain reference values are 0.0005, 0.001 and 0.0015, the trend of the experimental curves is the same as the simulated one. The upward of curve means the number of crack elements increases, and the downward means decreases.

Also, the peak values of the two curves occur approximately at the same strain level. When the reference strain is 0.0005, the peak strain of curves is 0.0018. The peak strain is 0.002 when the reference strain is 0.001 and 0.0015.

In conclusion, the simulated strain ratio curves are consistent with the experimental strain ratio curve, which indicates that the failure process of the simulation is in good agreement with the experimental one. The conclusion shows good consistency in simulation and experiment.

## 5. Conclusions

2D meso-scale FE models with real aggregates are developed using images obtained by digital image processing to simulate crack propagation processes in concrete under uniaxial compression loading. Cohesive elements with traction–separation laws are used to simulate complex nonlinear fracture. During the experiment, digital image correlation (DIC) was used to obtain the deformation and cracks of the specimens at different loading stages. It can be concluded that the fracture behaviour of concrete specimens is mainly shearing by analyzing the simulation results as well as strain contours and displacement contours obtained by DIC technology. The variation of stress contours demonstrates that the initial damage of ITZs results in a concentration of tensile stress in the vicinity region, which causes the initial crack. The downward section of the strain ratio curve indicates the existence of cracks healing phenomenon.

The high consistency between the simulation results and the experimental results indicates the feasibility of adopting a cohesive zone model in concrete fracture simulation under compression and shows the excellent performance of the realistic aggregate FE model with cohesive elements.

The crack propagation processes were obtained respectively by experimental strain contours and simulation, which are not entirely consistent. The experimental strain contours show that the crack appears before reaching the peak point of the curve. However, in the simulation, the first crack appears at the peak point of the curve. This is because the fracture process obtained by two-dimensional numerical simulation shows the inside fracture of 3D concrete. However, the cracks shown by DIC contours are those on the surface of three-dimensional specimens. There are voids and flaws inside the real 3D specimens, which also have an influence on cracking. So, these two results would not be totally consistent. That means 2D simulation cannot reveal the whole fracture of 3D concrete. That shows the limitation of 2D simulation.

## Figures and Tables

**Figure 1 materials-14-04314-f001:**
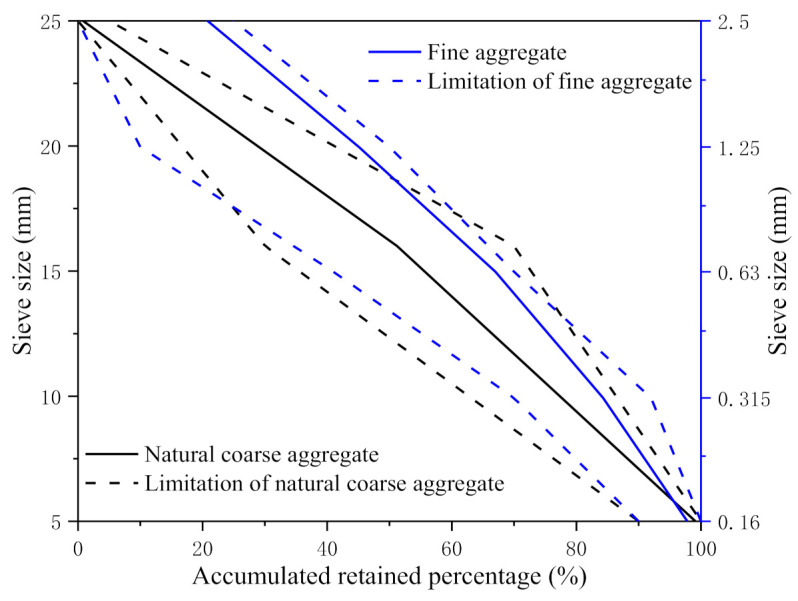
Aggregate grading curves.

**Figure 2 materials-14-04314-f002:**
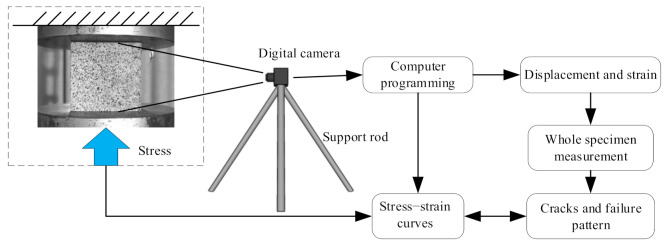
Digital image correlation (DIC) programming.

**Figure 3 materials-14-04314-f003:**
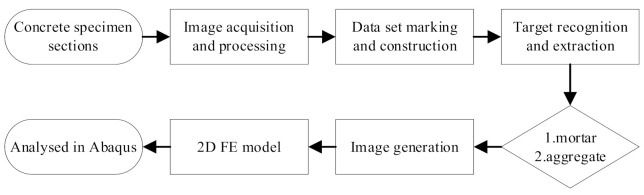
Digital image processing (DIP) procedure.

**Figure 4 materials-14-04314-f004:**
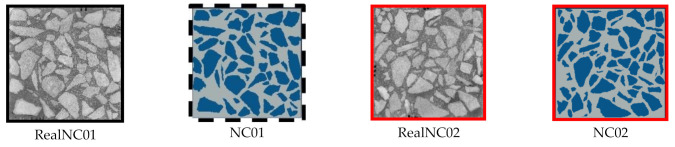
Real images and corresponding models.

**Figure 5 materials-14-04314-f005:**
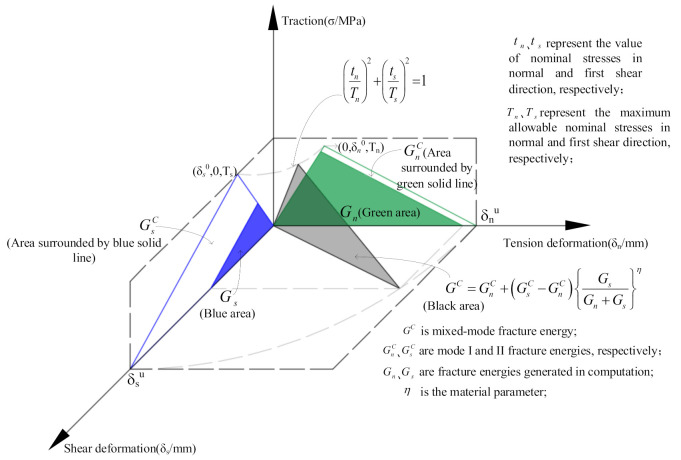
The quadratic nominal stress criterion and mixed-mode fracture criterion.

**Figure 6 materials-14-04314-f006:**
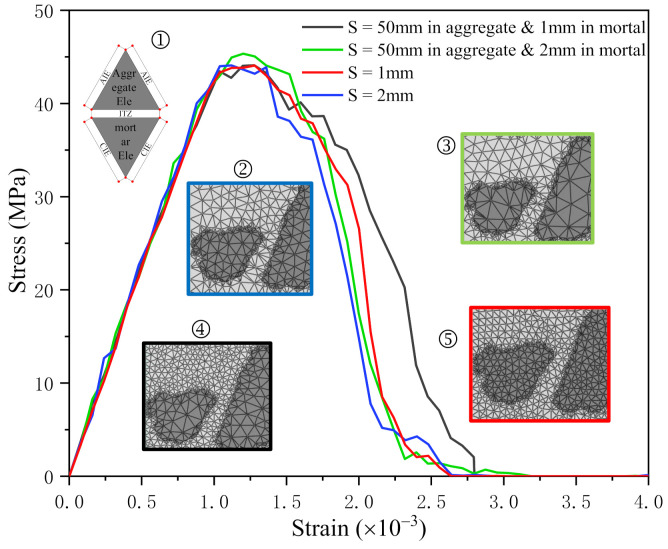
Stress−strain curves of different mesh forms: ➀ is cohesive element; ➁−➄ are different size of mesh forms.

**Figure 7 materials-14-04314-f007:**
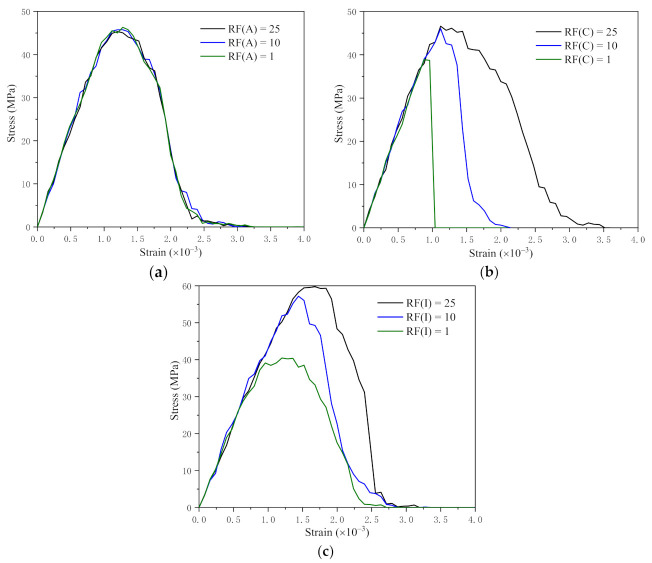
Different fracture energy ratios (RF) of three internal interfaces (AIE, CIE and ITZ): (**a**) Different fracture energy ratios of AIEs; (**b**) Different fracture energy ratios of CIEs; (**c**) Different fracture energy ratios of ITZs.

**Figure 8 materials-14-04314-f008:**
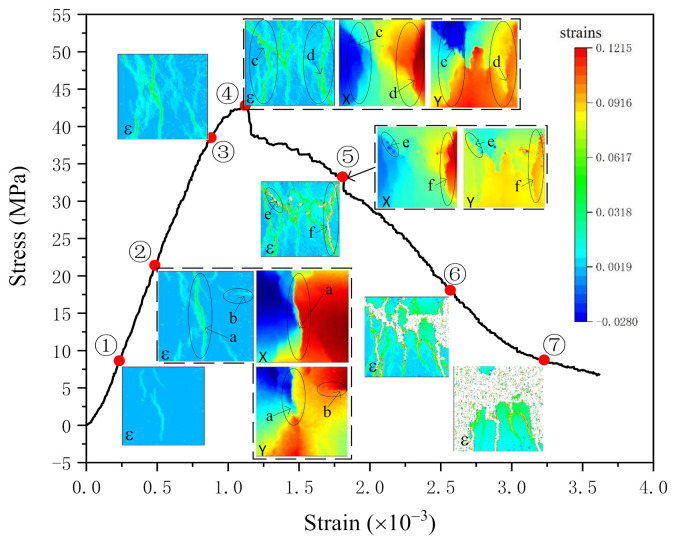
Strain and displacement contours of the experiment: ➀−➇ are the contours at different load stages; a−f are different cracks.

**Figure 9 materials-14-04314-f009:**
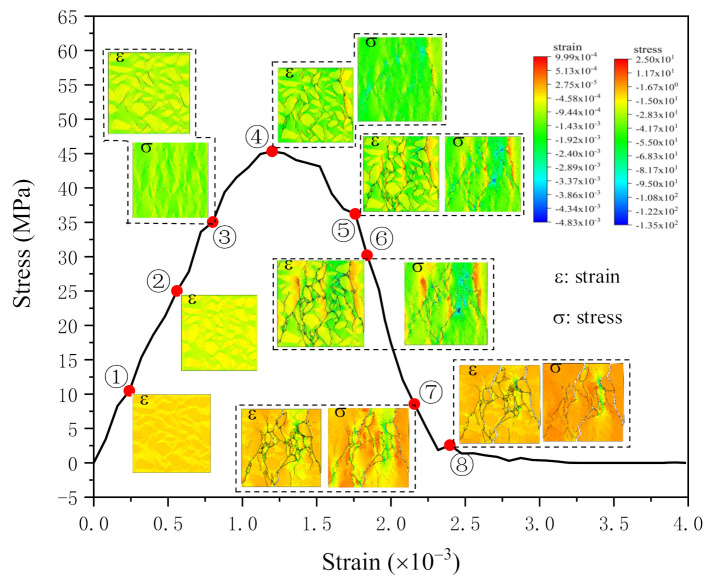
Strain and stress contours of NC01: ➀−➇ are the contours at different load stages.

**Figure 10 materials-14-04314-f010:**
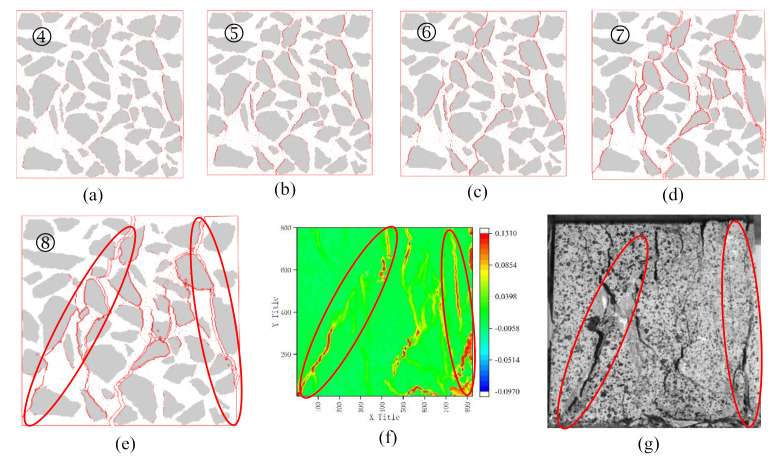
The cracks development process of NC01 and final cracks distribution of RealNC01: (**a**) The initial cracks appeared in ITZs; (**b**) Forming crack zone surround aggregates; (**c**) Cracks appeared in CIEs; (**d**) Forming the shear bands; (**e**) Final cracks distribution of NC01; (**f**) Final cracks distribution in contour of RealNC01; (**g**) Final cracks distribution of RealNC01; ➃−➇ corresponding to the point in Figure 9.

**Figure 11 materials-14-04314-f011:**
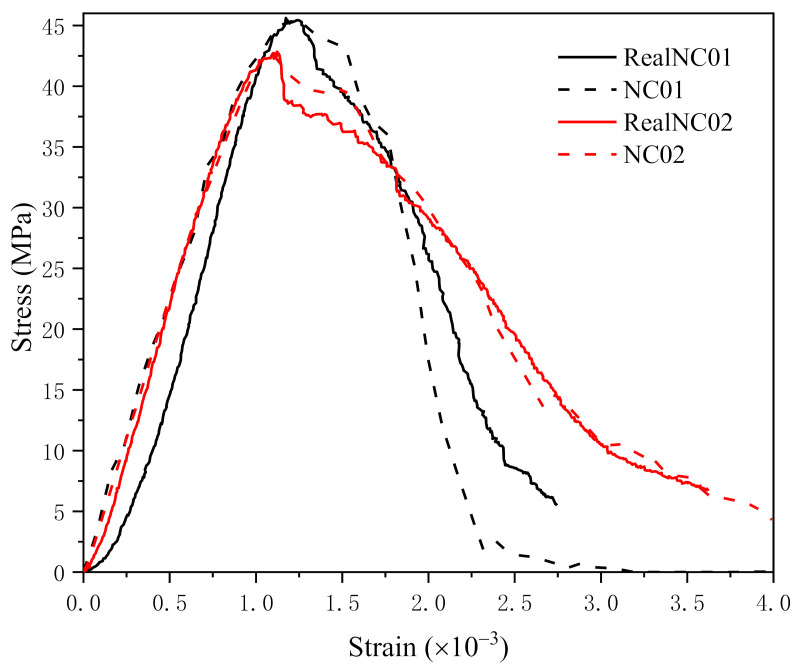
Stress−strain curves of experiment and simulation.

**Figure 12 materials-14-04314-f012:**
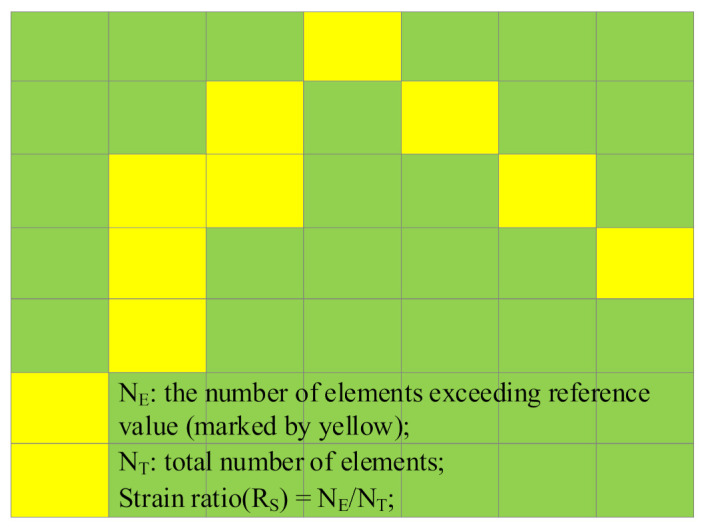
Sketch map of strain ratio.

**Figure 13 materials-14-04314-f013:**
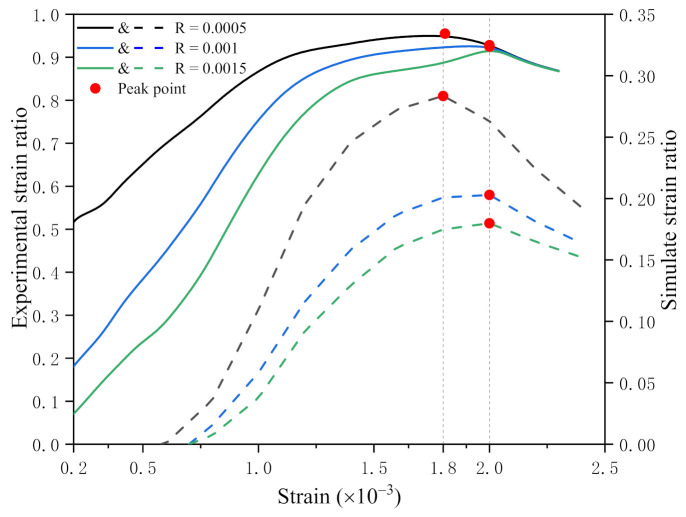
Strain ratio curves of different strain reference values.

**Table 1 materials-14-04314-t001:** Performance indexes of river sand.

Type	Fineness Modulus	Apparent Density (kg/m^3^)	Bulk Density (kg/m^3^)	Tight Density (kg/m^3^)	Water Absorption (%)	Dust Content (%)
FA	3.09	2620	1623	1718	0.59	0.8

**Table 2 materials-14-04314-t002:** Performance indexes of natural coarse aggregate.

Type	Apparent Density (kg/m^3^)	Bulk Density (kg/m^3^)	Crushing Index (%)	Water Absorption (%)	Dust Content (%)	Elongated and Flaky Particle (%)
NCA	2708	1436	12.37	0.39	1.12	7.1

**Table 3 materials-14-04314-t003:** Mix ratio of specimens.

No.	W/C	Water (kg/m^3^)	Cement (kg/m^3^)	Fine Aggregate (kg/m^3^)	Coarse Aggregate (kg/m^3^)
NC	0.5	203	406	730	1046

**Table 4 materials-14-04314-t004:** Material properties of different components.

Parts	Density (kg/m^3^)	Elastic Modulus (GPa)	Poisson’s Ratio	Maximum Nominal Stress in Normal/Shear Direction (MPa)	Normal/Shear Mode Fracture Energy (N/mm)	B-K Criterion Material Parameter
Aggregate	2708	75	0.16	-	-	-
Mortar	2400	35	0.20	-	-	-
AIEs	2600	10^3^	-	10/90	400/10,000	1.2
CIEs	2400	10^3^	-	5/35	0.2/4	1.2
ITZs	2300	10^3^	-	3/12	0.04/0.12	1.2

AIEs: interface elements inside aggregate; CIEs: interface elements inside cement paste; ITZs: Interface transition zone.

**Table 5 materials-14-04314-t005:** The number of solid elements (CPS3) and cohesive elements (COH2D4).

Elements Size (mm)	Solid Element (CPS3)	Cohesive Element (COH2D4)
1	30,624	40,496
2	19,603	26,901
50 in aggregate, 1 in cement paste	24,341	31,442
50 in aggregate, 2 in cement paste	17,033	25,441

## Data Availability

All the data is available within the manuscript.

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
