# Peer review of "Fracture Behaviour of Real Coarse Aggregate Distributed Concrete under Uniaxial Compressive Load Based on Cohesive Zone Model"

_materials, 2021, doi:10.3390/ma14154314_

Round 1

Reviewer 1 Report

This research provides valuable information regarding the computational modeling of concrete fracture. The subject is interesting and up-to-date.  However, certain improvements are necessary to publish the submitted manuscript. 

Introduction: The authors' classification does not address the computational models relying on the discrete element method (DEM). DEM-based micro and mesoscale fracture simulations (circular- and block-based representation) have been an active field of research since the early 2000s. Hence, to provide the big picture for the readers, the authors should refer to DEM developments in several sentences (not too long) and give the following most recent studies as references.

DOI: 10.1007/s10035-015-0546-4; DOI: 10.1016/j.engfracmech.2020.107269; DOI: 10.1007/s40571-020-00342-5, DOI: 10.1016/j.compgeo.2019.103290; DOI: 10.1002/suco.201100036; DOI: 10.1007/s40571-015-0044-9,

DOI: 10.1007/s10704-019-00373-x; DOI: 10.1016/j.ijrmms.2017.11.010,

DOI: 10.1016/j.cemconcomp.2021.104005

3. Simulating Progam: The authors may provide more info regarding digital image processing. It is on the air for readers. At least, the workflow may be briefly explained.

The element codes (e.g., CPS3) belongs to which software; please mention before using symbolic expressions.

3.2 Material Properties: Mode-III fracture is out-of-plane shearing. The presented research is a 2D model, so there is no way to model mode-III; hence, it should not be considered or added to the damage. The authors must better explain this part; otherwise, it is not clear what happened. Furthermore, how did the authors determine?

3.3 Mesh forms and boundary condition: Computational procedure was fully dynamic, or artificial damping was used to obtain quasi-static solutions. Please provide more info regarding the numerical procedure.

Figure 9. Instead of "Realistic," it is better to write "Real."

Page 14-Paragraph 3, there are some missing letters and figure numbers. Please read the revised version carefully.

Page 14-Paragraph 6, the peak values are not the same; they only occur approximately at the same strain level. Revise this paragraph as well.

General: The authors should also discuss the influence of voids and flaws on the strain evolution in the specimen. This may be the reason to observe the cracks before the peak in the experiment? Furthermore, the calibration procedure should be explained in the article. 

Author Response

Review report 1

This research provides valuable information regarding the computational modeling of concrete fracture. The subject is interesting and up-to-date. However, certain improvements are necessary to publish the submitted manuscript.

Reply: Thank you for your comments. I will answer your questions one by one and mark them in blue.

Comment 1: The authors' classification does not address the computational models relying on the discrete element method (DEM). DEM-based micro and mesoscale fracture simulations (circular- and block-based representation) have been an active field of research since the early 2000s. Hence, to provide the big picture for the readers, the authors should refer to DEM developments in several sentences (not too long) and give the following most recent studies as references.

Reply: Thank you for your comments. I have added the introduction of DEM in the introduction and cited those documents.

Comment 2: Simulating Progam: The authors may provide more info regarding digital image processing. It is on the air for readers. At least, the workflow may be briefly explained.

Reply: Thank you for your comments. I have added the flow chart of digital image processing (Figure 3).

Comment 3: The element codes (e.g., CPS3) belongs to which software; please mention before using symbolic expressions.

Reply: Thank you for your comments. Abaqus dynamic/explicit is used to do the calculation in this article. And it is mentioned in the article now.

Comment 4: 3.2 Material Properties: Mode-III fracture is out-of-plane shearing. The presented research is a 2D model, so there is no way to model mode-III; hence, it should not be considered or added to the damage. The authors must better explain this part; otherwise, it is not clear what happened. Furthermore, how did the authors determine?

Reply: Thank you for your comments. I made a mistake about the model. And I have revised it into we consider the value of both shear mode (mode II and mode III) is the same.

Comment 5: 3.3 Mesh forms and boundary condition: Computational procedure was fully dynamic, or artificial damping was used to obtain quasi-static solutions. Please provide more info regarding the numerical procedure.

Reply: Thank you for your comments. Abaqus dynamic/explicit is used to simulate concrete compression loading process. And the boundary conditions, loading, and step time are mentioned in the last paragraph of 3.3.

Comment 6: Figure 9. Instead of "Realistic," it is better to write "Real."

Reply: Thank you for your comments. I have revised it in the article.

Comment 7: Page 14-Paragraph 3, there are some missing letters and figure numbers. Please read the revised version carefully.

Reply: Thank you for your comments. I have revised it in the article.

Comment 8: Page 14-Paragraph 6, the peak values are not the same; they only occur approximately at the same strain level. Revise this paragraph as well.

Reply: Thank you for your comments. I have revised the description in the article, which should be described as you said.

Comment 8: The authors should also discuss the influence of voids and flaws on the strain evolution in the specimen. This may be the reason to observe the cracks before the peak in the experiment? Furthermore, the calibration procedure should be explained in the article.

Reply: Thank you for your comments. Those contents have been added in the article. And the section 3.4 shows the calibration of the fracture energy.

Reviewer 2 Report

The originality and the scientific value of the subject research are good.

The research area is fracture behavior of real coarse aggregate distributed concrete under uniaxial compressive load based on cohesive zone model.

The solved area is very interesting and also current.
The research includes an experimental program and numerical modeling.

The manuscript has the usual structure, but part of the discussion is missing.

The overall similarity of the calculation and the informative value must be substantially improved. 

It is necessary to provide more detailed information about the computational model,  parameters of the solver, and boundary conditions. 
It would be appropriate to take more account of the stochastic nature of the material properties of concrete (binder, aggregate, ITZ).

How the calculation affects the simplification of 3D to 2D task? 
Not sufficiently substantiated.

Indicate in a table all the important input parameters used in the calculation and solver FEM.

Provide more (graphical) outputs from the calculation. 

Overall, more calculations should be made to clarify the sensitivity of the model and the effect of the input parameters for the material models (concrete) and its parts.

Were tests and modeling of bigger structural elements done?

Overall, it is necessary to process the manuscript with greater interest.

Extensive research is underway in the area of concrete structures and nonlinear calculations of concrete structures when it is necessary to rework and expand the information in the introduction section. 
These are mainly the possibilities of material models of concrete, approaches to the choice of parameters, or taking into account the uncertainties in the calculation or stochastic character of concrete.

Sucharda, O., et. al. Numerical modelling and bearing capacity of reinforced concrete beams. Key Engineering Materials 2014, 577-578,  281-284.

Yu, Q. et. al. Numerical Study of Concrete Dynamic Splitting Based on 3D Realistic Aggregate Mesoscopic Model. Materials 2021, 14, 1948. 

The discussion chapter must be presented separately and present the results in the context of current research. What is the same, what is different?

Overall, it is necessary to improve the presentation of the results of experiments, numerical modeling and increase the informative value of the results.

The manuscript must be revised.

Author Response

Review report 2

Comment 1: The originality and the scientific value of the subject research are good. The research area is fracture behaviour of real coarse aggregate distributed concrete under uniaxial compressive load based on cohesive zone model. The solved area is very interesting and also current. The research includes an experimental program and numerical modeling. The manuscript has the usual structure, but part of the discussion is missing. The overall similarity of the calculation and the informative value must be substantially improved.

Reply: Thank you for your comments. I will answer your questions one by one and mark them in blue.

Comment 2: It is necessary to provide more detailed information about the computational model, parameters of the solver, and boundary conditions.

Reply: Thank you for your comments. The computation model used in this article is the quadratic nominal stress criterion and mixed-mode fracture criterion, it is showed in figure 5. And abaqus dynamic/explicit is used to simulate concrete compression loading process. And the boundary conditions, loading, and step time are mentioned in the last paragraph of 3.3.

Comment 3: It would be appropriate to take more account of the stochastic nature of the material properties of concrete (binder, aggregate, ITZ).

Reply: Thank you for your comments. The section 3.4 has been added to discuss the influence of fracture energy in interface elements.

Comment 4: How the calculation affects the simplification of 3D to 2D task?

Reply: Thank you for your comments. In my opinion. The fracture process obtained by two-dimensional numerical simulation show the inside fracture of 3D concrete. However, the cracks shown by DIC contours are those on the surface of three-dimensional specimens. And there are voids and flaws inside the real 3D specimens, which also have influence on cracking. So, these two results would not be totally consistent. That means 2D simulation cannot reveal the whole fracture of 3D concrete. That shows the limitation of 2D simulation.

Comment 5: Indicate in a table all the important input parameters used in the calculation and solver FEM. Provide more (graphical) outputs from the calculation.

Reply: Thank you for your comments. Those contents have been added in the article.

Comment 6: Overall, more calculations should be made to clarify the sensitivity of the model and the effect of the input parameters for the material models (concrete) and its parts.

Reply: Thank you for your comments. The mesh sensitivity is discussed in section 3.3. And the fracture energy discussion is added in section 3.4.

Comment 7: Were tests and modeling of bigger structural elements done?

Reply: Not yet, but the experience is under way.

Round 2

Reviewer 1 Report

In general, the revision of the manuscript looks ok; however, the reviewer still has two issues regarding this article. First, mode-III fracture energy should not be considered in the 2D model. The authors keep saying they have used the same value for modes II and III, but there is no mode-III. In Figure 5, what is the second shear direction. In 2D, there, you can only slide on a line, not a surface. Hence, this point should be clarified and corrected. Also, there were some misleading definitions in the annotation table, like stress - it is "stress tensor". Please update the rest accordingly. Secondly, the English should be improved, and proofread is required before publication.

Author Response

Comment: In general, the revision of the manuscript looks ok; however, the reviewer still has two issues regarding this article. First, mode-III fracture energy should not be considered in the 2D model. The authors keep saying they have used the same value for modes II and III, but there is no mode-III. In Figure 5, what is the second shear direction. In 2D, there, you can only slide on a line, not a surface. Hence, this point should be clarified and corrected. Also, there were some misleading definitions in the annotation table, like stress - it is "stress tensor". Please update the rest accordingly. Secondly, the English should be improved, and proofread is required before publication.

Reply: Thank you for your comments. Your corrections solved my doubts about this model and made me understand this model better. Thanks again for your correction and patience. And the other improvements also have been made in the article.

Reviewer 2 Report

Thank you for the adjustments made, but the answers are not sufficiently adequate.

The changes made the improvement of the manuscript.

It is clear from the comments that the authors understand the issue. However, the manuscript does not clearly define the research issue and what will be a new knowledge. 

The modifications of the manuscript made are partial.
Extensive research is devoted to the issue of numerical simulations of concrete and it is necessary to put the chosen approach into context.

I would encourage the authors to revise the manuscript with a greater focus on presenting new knowledge and benefits for further research. Numerical modeling itself has already been performed and presented in an interesting way. 
Focus (rewrite and extended) mainly on the Introduction, Result analysis (Discussion) and Conclusion parts.

Author Response

Comment 1: It is clear from the comments that the authors understand the issue. However, the manuscript does not clearly define the research issue and what will be a new knowledge. 

Reply: Thank you for your comments and patience. This part has been added to the introduction. The objective of this study is to provide a new idea for studying meso-scale compression failure of concrete.

Comment 2: The modifications of the manuscript made are partial.
Extensive research is devoted to the issue of numerical simulations of concrete and it is necessary to put the chosen approach into context.

Reply: Thank you for your comments. The third paragraph of the introduction is added to introduce the cohesive zone model used in this study.

Comment 3: I would encourage the authors to revise the manuscript with a greater focus on presenting new knowledge and benefits for further research. Numerical modeling itself has already been performed and presented in an interesting way.

Reply: Thank you for your comments. Those contents have been added in the introduction and conclusion.

Round 3

Reviewer 1 Report

In Figure 5, the authors should delete tt since it is for mode-III. 

Author Response

Comment: In Figure 5, the authors should delete tt since it is for mode-III.

Reply: Thank you very much for your comment. The content has been modified.

Reviewer 2 Report

The research can be better understood from the manuscript.
The manuscript has sufficient information value.

Author Response

Thank you very much!